# Transcriptome Analysis Reveals Differentially Expressed Genes Involved in Cadmium and Arsenic Accumulation in Tea Plant (*Camellia sinensis*)

**DOI:** 10.3390/plants12051182

**Published:** 2023-03-05

**Authors:** Shiqi Liu, Xuqian Peng, Xiaojing Wang, Weibing Zhuang

**Affiliations:** 1College of Tea Science, Guizhou University, Guiyang 550025, China; 2Jiangsu Key Laboratory for the Research and Utilization of Plant Resources, Institute of Botany, Jiangsu Province and Chinese Academy of Sciences, Nanjing Botanical Garden Mem. Sun Yat-Sen, Nanjing 210014, China

**Keywords:** transcriptome, Cd and As, tea plants, RNA-sequencing, WGCNA

## Abstract

Tea (*Camellia sinensis*) is the second most consumed drink in the world. Rapid industrialization has caused various impacts on nature and increased pollution by heavy metals. However, the molecular mechanisms of cadmium (Cd) and arsenic (As) tolerance and accumulation in tea plants are poorly understood. The present study focused on the effects of heavy metals Cd and As on tea plants. Transcriptomic regulation of tea roots after Cd and As exposure was analyzed to explore the candidate genes involved in Cd and As tolerance and accumulation. In total, 2087, 1029, 1707, and 366 differentially expressed genes (DEGs) were obtained in Cd1 (with Cd treatment for 10 days) vs. CK (without Cd treatment), Cd2 (with Cd treatment for 15 days) vs. CK, As1 (with As treatment for 10 days) vs. CK (without Cd treatment), and As2 (with As treatment for 15 days) vs. CK, respectively. Analysis of DEGs showed that a total of 45 DEGs with the same expression patterns were identified in four pairwise comparison groups. One ERF transcription factor (CSS0000647) and six structural genes (CSS0033791, CSS0050491, CSS0001107, CSS0019367, CSS0006162, and CSS0035212) were only increased at 15 d of Cd and As treatments. Using weighted gene co-expression network analysis (WGCNA) revealed that the transcription factor (CSS0000647) was positively correlated with five structural genes (CSS0001107, CSS0019367, CSS0006162, CSS0033791, and CSS0035212). Moreover, one gene (CSS0004428) was significantly upregulated in both Cd and As treatments, suggesting that these genes might play important roles in enhancing the tolerance to Cd and As stresses. These results provide candidate genes to enhance multi-metal tolerance through the genetic engineering technology.

## 1. Introduction

Environmental pollution with heavy metals has great threats to human health and our living environment [1]. The high speed of industrialization causes heavy metal pollution around industrial enterprises in many countries [2,3,4]. The presence of heavy metals in waste of many industries has attracted much attention due to their toxicity to many life forms [5]. Heavy metal stress has notable adverse effects on crop productivity and growth, including growth inhibition, low photosynthesis, reduction the accumulation of biomass, senescence, chlorosis, and even plant death [6,7].

Heavy metals, including cadmium (Cd), arsenic (As), chromium (Cr), silver (Ag), lead (Pb), nickel (Ni), cobalt (Co), zinc (Zn), and iron (Fe), are important environmental pollutants, which are responsible for immense damage in plant growth and metabolism [8]. The most common and dangerous metal pollutants Cd and As, which exist simultaneously in cultivated soil [9]. In plants, Cd toxicity caused the easily identifiable symptoms of chlorosis and shunted growth, and excessive Cd in plants usually inhibited the plant growth and development and even caused necrosis [10,11]. Cd contamination inhibited the rate of CO_2_ fixation, decreased chlorophyll (chl) content, and depressed photosynthetic activity [12,13]. Plants grown on Cd-contaminated soil are subjected to osmotic stress by minimizing leaf relative water content, reducing transpiration rate, and stomatal conductance [14]. Cd toxicity caused the accumulation of reactive oxygen species (ROS) that damaged to cellular membranes, important cell biomolecules (proteins, DNA, and lipids), and organelles [15,16]. Moreover, Cd can also decrease the Fe and Zn uptake, reduce the Fe and Zn concentrations, and cause more severe leaf chlorosis [17,18,19,20]. Jiang et al. [21] revealed that Cd interacted with the mineral nutrients (Ca, Mn, Mg, K, and P). Arsenic (As), a toxic metalloid that is ranked 20th in natural abundance, is widely distributed in the environment [22]. Exposure to As (V) can cause numerous stress in plants, such as growth inhibition and several physiological disorders of plants, and can finally lead to death [23,24]. Arsenic toxicity induces the production and accumulation of ROS that damage biomolecules (proteins, DNA, and lipids) and eventually cause cell death [25,26].

Transcriptome analysis was widely used to investigate the molecular mechanism of plant response to heavy metal stress [27]. To date, the differentially expressed genes (DEGs) following Cd and As exposure were identified by transcriptome analysis in many plants, including *Arabidopsis* [28], *C*. *sinensis* [29], rice [30,31], *tall fescue* [32], and *phytolacca americana* [33]. Among these differentially expressed genes, several structural gene families, including ATP-binding cassette (ABC) transporters, zinc/iron-regulated transporter-like proteins (ZIPs), heavy-metal ATPases (-HMAs), galactinol synthase (GolS), 9-cis-epoxycarotenoid dioxygenase (-NCED), cation diffusion facilitators (CDFs), natural resistance-associated macrophage proteins (Nramps), and heavy-metal associated isoprenylated plant proteins (HIPPs) [34,35,36,37], are involved in the detoxification in plants growing in metal-rich soil. Moreover, several transcription factor (TF) families such as Mitogen-activated protein kinase (MAPK), APETALA2/ethylene responsive element binding factors (AP2/ERF), Myb avian myeloblastosis viral oncogene (MYB), WRKY, and basic-region leucine zipper (bZIP) have been reported to be involved in the regulation of heavy metal tolerance and detoxification in many plants such as *Arabidopsis*, rice, and barley [38,39,40,41,42,43].

Due to the difference in chemical properties of As and Cd, concurrent minimizing their uptake poses a problem. In the present study, we first investigated the effect of Cd and As on plant phenotype, antioxidant enzyme activities, and molecular regulation mechanisms of the tea plants. Comparative transcriptome analysis of Cd/As-treated and non-treated tea roots was performed to identify the DEGs. A clear and detailed view of the transcriptomic changes triggered by Cd and As exposure is important to understand the gene expression network of the basal response to Cd and As stresses. We identified eight DEGs with the same expression patterns after Cd and As treatments, including two transcription factors and six structural genes by performing pairwise comparative analysis. Through this study, we hope to identify candidate genes to enhance multi-metal tolerance through the genetic engineering technology.

## 2. Results

### 2.1. Determination of Antioxidant Enzyme Activities and MDA Content of Cd/As-Treated Tea Roots

To explore the effect of Cd and As on plant growth and development, the phenotypic changes in tea plants after Cd and As treatments were investigated, respectively. In this study, the number of lateral roots was significantly reduced in tea seedlings treated with 120 mg/kg As (II) [44] solution for 15 days in comparison with the control group (CK), and the roots turned brown in tea seedlings treated with 120 mg/kg Cd solution for 15 days in comparison with the control group. However, there were no significant differences between the control group and Cd/As-treated tea roots for 10 days (Figure 1). Moreover, we further investigate the effect of heavy metal Cd and As on antioxidant enzyme activities and malondialdehyde (MDA) content, the activities of superoxide dismutase (SOD), catalase (CAT), and peroxidase (POD), and MDA content in Cd/As-treated tea roots. Our results showed that the SOD level was significantly decreased in Cd/As-treated tea roots in comparison with the control group, while the MDA level was significantly enhanced in Cd/As treated-tea plants. Although the CAT and POD levels were slightly increased in Cd/As treated-tea plants in comparison with the control group, there is no significant difference in CAT and POD content between Cd/As-treated tea plants and control groups.

### 2.2. Transcriptome Analysis of the Cd/As-Treated and Non-Treated Tea Roots

To further investigate the molecular regulatory mechanisms of tea plants response to heavy metals Cd and As stresses, fifteen high-quality cDNA libraries were constructed and used to perform high throughput RNA-seq (Table 1). As shown in Table 1, 649,096,012 raw reads ranging from 39,422,132 to 46,710,062 for each library and 634,712,730 clean reads ranging from 38,261,120 to 45,841,500 for each library were obtained. The Q30 percentage and average GC content were over 90% and 44.15~45.24% for the libraries, respectively. Moreover, 75.42–83.8% of clean reads were assembled into the genome of *C. sinensis* for each library. Thus, a total of 39,153, 38,538, 39,869, 36,481, 39,746, 39,366, 39,580, 39,212, 38,744, 37,658, 35,675, 37,408, 39,304, 38,725, and 59,129 genes for each library were identified in this study (Table 1). Thus, these results further demonstrated that RNA-seq data are of high quality and suitable for subsequent analyses. In addition, expression levels of the transcripts were calculated based on the FPKM (fragments per kilobase of exon per million fragments mapped) normalization method using the DESeq2 package. Correlation analysis showed that the 15 root samples of tea plants were clustered into five different groups, and the different replicates in the same group processed a strong positive correlation (*r*  > 0.89), indicating that there was high reproducibility and reliability in transcriptome data (Figure 2A and Appendix A). A volcano plot analysis was performed to identify the DEGs between the control group (CK) and Cd/As-treated tea roots for 10 d (named Cd1 and As1, respectively) and 15 d (named Cd2 and As2, respectively): Cd1 vs. CK, Cd2 vs. CK, As1 vs. CK, and As2 vs. CK (Appendix A). Thus, a total of 2,087 DEGs (319 up- and 1768 down-regulated transcripts), 1029 DEGs (345 up- and 864 down-regulated transcripts), 1707 DEGs (309 up- and 1398 down-regulated transcripts), and 366 DEGs (74 up- and 292 down-regulated transcripts) were detected in Cd1 vs. CK, Cd2 vs. CK, As1 vs. CK, and As2 vs. CK, respectively. A Venn diagram showed that 552 DEGs were present in Cd1 vs. CK and Cd2 vs. CK, and 178 DEGs were distributed in As1 vs. CK and As2 vs. CK. In addition, fifty-three DEGs are simultaneously distributed in the four comparison groups: Cd1 vs. CK, Cd2 vs. CK, As1 vs. CK, and As2 vs. CK, implying that these genes may be simultaneously involved in the regulation of heavy metal Cd and As stresses.

### 2.3. GO and KEGG Enrichment Analyses of DEGs

GO enrichment analysis was performed to identify the biological function of DEGs using Blast2GO (Figure 3 and Appendix A). Numbers of 1091 DEGs (Cd1 vs. CK), 637 DEGs (Cd2 vs. CK), 945 DEGs (As1 vs. CK), and 221 DEGs (As2 vs. CK) were clustered into 158, 129, 99, and 60 functional groups, respectively (Appendix A). Moreover, the top 25 enriched GO terms of the DEGs in different pairwise comparisons: Cd1 vs. CK, Cd2 vs. CK, As1 vs. CK, and As2 vs. CK. Our results showed that the top 25 enriched GO terms of the DEGs in Cd1 vs. CK were divided into three GO categories, including five biological process categories, one cellular component category, and 19 molecular function (MF) categories. The five most enriched GO terms, including oxidoreductase activity (GO:0016491, 219 DEGs), cofactor binding (GO:0048037, 135 DEGs), catalytic activity (GO:0003824, 668 DEGs), heme binding (GO:0020037, 85 DEGs), and tetrapyrrole binding (GO:0046906, 85 DEGs), were distributed in MF terms. The top 25 significant items in the enrichment analysis of GO-term for the DEGs in Cd2 vs. CK were divided into three GO categories, including 14 biological process categories, one cellular component category, and 10 MF categories. The five most enriched MF terms were oxidoreductase activity (GO:0016491, 131 DEGs), membrane (GO:0016020, 94 DEGs), catalytic activity (GO:0003824, 396 DEGs), heme binding (GO:0020037, 46 DEGs), and tetrapyrrole binding (GO:0046906, 46 DEGs), respectively. Moreover, the top 25 significant items in the enrichment analysis of GO-term for the DEGs in As1 vs. CK were divided into three GO categories, including 14 biological process categories, one cellular component category, and 10 MF categories. The five most enriched MF terms were oxidoreductase activity (GO:0016491, 216 DEGs), cofactor binding (GO:0048037, 135 DEGs), catalytic activity (GO:0003824, 619 DEGs), heme binding (GO:0020037, 86 DEGs), and oxidation-reduction process (GO:0055114, 202 DEGs), respectively. In addition, the top 25 significant items in the enrichment analysis of GO-term for the DEGs in As1 vs. CK were divided into three GO categories, including one biological process category, four cellular component categories, and 20 MF categories. The five most enriched MF terms were oxidoreductase activity (GO:0016491, 53 DEGs), cofactor binding (GO:0048037, 33 DEGs), catalytic activity (GO:0003824, 167 DEGs), heme binding (GO:0020037, 22 DEGs), and tetrapyrrole binding (GO:0046906, 22 DEGs), respectively.

To further investigate the DEGs involved in the metabolic pathways, KEGG pathway analysis of DEGs identified in different pairwise comparisons was carried out (Appendix A). Numbers of 161 DEGs (Cd1 vs. CK), 80 DEGs (Cd2 vs. CK), 139 DEGs (As1 vs. CK), and 32 DEGs (As2 vs. CK) were assigned to 192, 101, 183, and 59 KEGG pathways, which were further divided into six categories such as metabolism, environmental information processing, organismal systems, human diseases, cellular processes, and genetic information processing (Appendix A). Among them, the top five metabolism pathways with the largest number of DEGs annotated by KEGG in four pairwise comparisons: Cd1 vs. CK, Cd2 vs. CK, As1 vs. CK, and As2 vs. CK (Figure 4A–D). Notably, the galactose metabolism pathway (ko00052) was commonly distributed in four pairwise comparisons, suggesting that this pathway played important roles in response to heavy metal stress.

### 2.4. Identification of Differentially Expressed TFs in Cd/As-Treated Tea Roots

It is well known that MAPK, AP2/ERF, MYB, WRKY, and bZIP TFs play important roles in response to heavy metal stress. In this study, the twelve family TFs with differential expressions were detected in this RNA-seq data (Figure 5 and Appendix A). Among them, 23 MYB, 33 AP2, 16 bHLH, 14 WRKY, 12 GRAS, six DOF, five bZIP, one GRF, one ZF-HD, one HSF, one SBP, and one WD40 were differentially expressed in Cd1 vs. CK. Numbers of 14 MYB, 27 AP2, seven bHLH, six WRKY, four Dof, four bZIP, three GRAS, one GRF, one ZF-HD, and one WD40 were differentially expressed in Cd2 vs. CK. Moreover, 15 MYB, 14 bHLH, 28 AP2, six Dof, four GRAS, three bZIP, two ARF, one WRKY, one GRF, one C3H, and one WD40 were also differentially expressed in As1 vs. CK. Four MYB, three bHLH, five AP2, three GRAS, one bZIP, one TCP, and one WD40 were differentially expressed in As2 vs. CK. Further analysis showed that one MYB TF (CSS0030465) was downregulated at both 10 d and 15 d under Cd and As treatments. Two AP2/ERF (CSS0018401 and CSS0049619) and one bHLH (CSS0041447) were increased to a maximum at 10 d and decreased sharply at 15 d of Cd and As treatments, which may be short-term response genes in *C. sinensis* under the Cd and As treatments. In addition, one AP2/ERF (CSS0000647) was only up-regulated at 15 d after Cd and As treatments, indicating that this gene may be slowly responsive to heavy metal Cd and As stresses.

### 2.5. Identification of DEGs in Cd/As-Treated Tea Roots

To further investigate the molecular regulation mechanisms of heavy metal Cd/As accumulation and detoxification, comparative transcriptome analysis was performed to identify the genes involved in the regulation of Cd and As stresses. In the present study, a total of 45 DEGs with differential expression patterns were identified in both Cd- and As-treated tea roots in comparison with the control group (Figure 6 and Appendix A). Among them, 82.2% (37) of 45 DEGs were down-regulated in four pairwise comparisons: Cd1 vs. CK, Cd2 vs. CK, As1 vs. CK, and As2 vs. CK, indicating that these DEGs played a negative regulatory role in response to heavy metal Cd and As stresses. Moreover, the expression levels of seven DEGs (CSS0000647, CSS0033791, CSS0050491, CSS0001107, CSS0019367, CSS0006162, and CSS0035212) were decreased to a minimum at 10 d then increased sharply at 15 d of Cd and As treatments. In addition, only one gene (CSS0004428) was up-regulated after 10 d and 15 d of Cd and As treatments, indicating that this gene was involved in enhancing tolerance to Cd and As stresses (Appendix A).

### 2.6. Regulation Network in Tea Plants under Heavy Metal Cd/As Stress

To further identify the key modules, hub genes, and possible molecular regulation mechanisms involved in response to Cd and As stresses, we performed WGCNA analysis using the 45 DEGs with the same expression patterns in four pairwise comparison groups (Figure 7 and Appendix A). Our results showed that 44 (97.8%) of 45 DEGs in both Cd and As-treated tea roots were analyzed together to construct the gene co-expression network analysis (Figure 7A,B). The ERF transcription factor (CSS0000647) is localized at the center of the network, which was positively correlated with five functional genes such as three *CsGolS* family genes (CSS0001107, CSS0019367, and CSS0035212), one *CsNCED* (CSS0033791), and one *CsHIPP* (CSS0006162), suggesting that the ERF transcription factor played positive roles in the transcriptional regulation/connection of -GolS*,* NCED, and HIPPgenes.

### 2.7. Validation of Expression Patterns of Eight Candidate DEGs Associated with Cd/As Stress Using qRT-PCR

To verify the accuracy of RNA-seq results, confirmation of expression levels of eight functional genes, including six structural genes and two transcription factors CSS0050491, CSS0019367, CSS0033791, CSS0001107, CSS0035212, CSS0006162, CSS0000647, and CSS0004428 were performed using qRT-PCR assays (Figure 8). The log_2_-fold change (FC) value of these genes was calculated to estimate the effect of the Cd/As treatment. As shown in Figure 8, the qRT-PCR results were generally consistent with the RNA sequencing results (Appendix A).

## 3. Discussion

The presence of heavy metals in agricultural soils is of major environmental concern and a great threat to life on the earth. Tea is the most popular beverage in the world after water. The absorption and accumulation of heavy metals (Pb, As, Cu, Cd, and Hg) in tea plants can enter the human body and increase the concentration of toxic metals in the human body [45,46]. Although the determination of some heavy metals in fresh and processed tea leaves has been investigated [47], the molecular regulation mechanisms of Cd and As stresses response in tea plants was still unclear. In the present study, the effect of Cd and As on plant growth and antioxidant enzyme activities was first investigated, and the results showed that the number of lateral roots was significantly reduced in Cd-treated seedlings, and the healthy white roots turned brown in As-treated seedlings. Moreover, the SOD level was significantly decreased in Cd/As treated-tea plants in comparison with the control group, while the MDA level was significantly enhanced in Cd/As treated-tea plants, and the results of Cd treatment was consistent with the previous studies [48].

In recent years, transcriptome profiling was widely used to explore the molecular regulation mechanism of many plants’ response to heavy metal stress. For example, Kintlová et al. [49] performed transcriptome analysis of barley under three different heavy metal (Zn, Cu, and Cd) stresses. Zhang et al. [50] investigated the molecular mechanism of soybean root responding to Cd stress using RNA-seq analysis, and the results further showed that three isoflavones 2′hydroxylase genes, two isoflavone reductase genes, and a chalcone synthase gene might be involved in regulating Cd stress. Transcriptome profiling was used to identify the genes and pathways associated with arsenic toxicity and tolerance in *Arabidopsis* [51]. Although the absorption and subcellular distribution of some heavy metals in tea plant have been investigated, the molecular mechanism underlying the enhanced multi-metal tolerance in tea plant remains unclear [46]. In this study, we obtained 634, 712, 730 high-quality reads from *C. sinensis* on the Illumina HiSeqTM 2000 platform and identified 598,588 genes which were identified by mapping to a reference genome.

The DEGs in Cd/As-treated tea roots in comparison with the control group were identified based on our RNA-seq data analysis. Numbers of 2087, 1029, 1707, and 366 DEGs were identified in four different pairwise comparison groups: Cd1 vs. CK, Cd2 vs. CK, As1 vs. CK, and As2 vs. CK, respectively. GO analysis further showed that 1,091 DEGs in Cd1 vs. CK, 637 DEGs in Cd2 vs. CK, 945 DEGs in As1 vs. CK, and 221 DEGs in As2 vs. CK were clustered into 158, 129, 99, and 60 functional groups, respectively, and the results further showed that the five most enriched MF terms were oxidoreductase activity, cofactor binding, catalytic activity, heme binding, and tetrapyrrole binding. Moreover, KEGG pathway analysis showed that 161 DEGs (Cd1 vs. CK), 80 DEGs (Cd2 vs. CK), 139 DEGs (As1 vs. CK), and 32 DEGs (As2 vs. CK) were assigned to 192, 101, 183, and 59 KEGG pathways, and the galactose metabolism pathway (ko00052) were commonly distributed in four pairwise comparisons, suggesting that this pathway played crucial roles in response to heavy metal stress.

It is well known that MAPK, AP2, ERF, MYB, WRKY, and bZIP TFs play important roles in response to heavy metal stress. Our results showed that twelve family TFs with differential expressions were detected in Cd/As-treated tea seedlings based on the RNA-seq data (Figure 5). Seven MYBs (CSS0030465, CSS0031744, CSS0000442, CSS0002706, CSS0006502, CSS0043569, and CSS0015964) were reduced in Cd-treated tea roots, suggesting that these genes play a negative regulatory role in response to Cd stress. Two MYB (CSS0030465 and CSS0032956) and three AP2 (CSS0034513, CSS0009309, and CSS0043971) were reduced in As-treated tea roots, suggesting that these TFs play a negative regulatory role in response to As stress. Homologous genes of these nine MYBs and three AP2 in *Arabidopsis* and other plants further confirmed that these genes were involved in regulating heavy metal tolerance [52,53,54,55,56].

Our results showed that a total of 45 DEGs with the same expression patterns were identified in four pairwise comparison groups. Among them, seven DEGs (CSS0000647, CSS0033791, CSS0050491, CSS0001107, CSS0019367, CSS0006162, and CSS0035212) were decreased to a minimum at 10 d then increased sharply at 15 d of Cd and As treatments, suggesting that these genes displayed less sensitivity to mild Cd and As stresses. Functional analysis showed that the orthologous genes of these genes were involved in response to various heavy metal stress [57,58,59,60]. For example, the orthologous genes of seven DEGs (CSS0000647, CSS0033791, CSS0050491, CSS0001107, CSS0019367, CSS0035212, and CSS0006162) in *Arabidopsis* were *AtERF53*, *AtNCED3*, *AtRFS5*, *AtHMP41,* and three *AtGolS* genes, respectively (Appendix A). Previous study had revealed that ectopic expression of *MhNCED3* in *Arabidopsis* enhanced the tolerance to Cd stress by increasing ABA level and alleviated Cd-induced cell death [58]. Ranjan et al. (2023) had revealed that overexpression of *AtGolS* gene in *Arabidopsis* promoted the galactinol accumulation and improved growth under As stress [60]. Previous study had revealed that overexpression of *AtHIPP06* and *AtHIPP26*conferred Cd tolerance to transgenic plants, whereas the triple knockout mutant *AtHIPP20/21/22* gained greater sensitivity to Cd than the wild-type of *Arabidopsis* [61]. ETHYLENE RESPONSE FACTOR 53 (*AtERF53*) belongs to group 1 in the ERF family and is induced in the early hours of dehydration and salt treatment [62,63]. The results of WGCNA analysis revealed that the ERF transcription factor (CSS0000647) was positively correlated with five functional genes such as three *CsGolS* genes (CSS0001107, CSS0019367, and CSS0006162), one *CsNCED* (CSS0033791), and one *HIPP* (CSS0006162), suggesting that the ERF transcription factor might also play important roles in enhancing the tolerance to Cd and As stresses. In addition, one *CsCBL* (CSS0004428) was up-regulated in both Cd and As treatments, indicating that these genes played important roles in enhancing the tolerance to Cd and As stresses in *C. sinensis* (Appendix A).

## 4. Materials and Methods

### 4.1. Plant Growth Conditions and Heavy Metal Treatments

New tea plants are propagated by Fuding Dabaicha (*C. sinensis* ‘Fuding Dabaicha’) purchased from Zhongfeng Tea Garden in Ya’an famous mountain area. Two-year-old seedlings were planted in pots (25 × 30 × 25 cm) and grown in the experimental field of Guizhou university. The roots of two-year-old untreated tea seedlings were collected and used as control samples. For Cd treatment, two-year-old tea seedlings were poured with 400 mL Cd solution with the Cd (II) concentration of 120 mg/kg. For As treatment, two-year-old tea seedlings were poured with 400 mL As solution with the As (V) concentration of 120 mg/kg. The tea roots were collected after 10 d and 15 d of Cd and As treatments, respectively. The root samples with three replicates were harvested and frozen in liquid nitrogen immediately and stored in a deep freezer at −80 °C for further study. Total RNA was extracted from the different root samples using TRIzol reagent (Invitrogen, Carlsbad, CA, USA) following the manufacturer’s protocol. The total RNA for each sample was analyzed on an Agilent 2100 Bioanalyzer with RNA 6000 Nano Labchip kit components to determine the integrities and concentration of the tea plant RNA samples.

### 4.2. Library Construction and RNA-Seq

The mRNA was purified from total RNA using poly-T oligo attached magnetic beads. After purification, the mRNA was cleaved into small fragments of 120~210 bp using divalent cations at an elevated temperature. Then, the cleaved RNA fragments were constructed into the final cDNA library based on the protocol for the mRNA-Seq sample preparation kit (Illumina, San Diego, CA, USA). The average insert size for the paired-end libraries was 300 bp (±50 bp), and the paired-end sequencing was performed on an Illumina Hiseq2000/2500 (LC Sciences, Houston, TX, USA) following the vendor’s recommended protocol. Eighteen RNA libraries included six control libraries, six Cd-treated libraries, and six As-treated libraries.

### 4.3. Sequencing Analysis, Transcripts Assembly, and Functional Annotation

After sequencing, raw reads were trimmed by removing Illumina adapters sequences and low-quality bases. The high-quality clean reads were mapped to the *C. sinensis* genome (reference genome) using HISAT (Hierarchical indexing for spliced alignment of transcripts). The gene expression was calculated and normalized to a Reads Per Kilobase per Million clean reads (FPKM) value. The false discovery rate (FDR) was calculated by SAS 8.0, which was used to adjust the *p*-value threshold. Those genes with FDR < 0.05 and |log_2_Fold Change| > 1 were considered DEGs.

For functional annotation, seven databases such as NR, NT, Pfam, KOG, Swiss-Prot, KEGG, and GO were used to annotate the assembled genes of *C. sinensis* using the BLASTX program.

### 4.4. qRT-PCR Validation of Candidate Genes Identified in RNA-Seq

To verify the results of RNA-Seq data, eight candidate DEGs were selected and performed qRT-PCR analysis. The housekeeping *CsGAPDH*gene was selected as a reference for normalization in qRT-PCR analysis [64]. The 2^−ΔΔCT^ method was performed to calculate relative expression changes in selected genes. Relative expression values were obtained from three biological repeats and measured for three technical repeats. The PCR primers of eight candidate genes are shown in Appendix A.

### 4.5. Regulation Network in Tea Plants under Heavy Metal Cd/As Stress

The WGCNA analysis was conducted to identify the key modules, hub genes, and possible molecular mechanisms of Cd and As stresses based on the FPKM values of the DEGs identified in four pairwise comparison groups with the help of an R package (v1.68) [65]. CYTOSCAPE (v3.7.1) was then used to visualize the networks of genes within the module and to present the biological interaction of core genes [66].

### 4.6. Statistical Analysis

Data were statistically analyzed using IBM^®^ SPSS^®^ Statistics 20 (IBM, USA). Analysis of variance (ANOVA) and mean separation were performed using *t*-test or one-way ANOVA with the least significant difference (LSD) at *p* < 0.05.

## 5. Conclusions

In this study, a large number of DEGs were obtained from RNA-seq data of Cd/As-treated tea roots, which provided more opportunities for studying the molecular regulation mechanisms of Cd and As stresses response in tea plants. A total of 2087, 1029, 1707, and 366 DEGs were identified in Cd1 vs. CK, Cd2 vs. CK, As1 vs. CK, and As2 vs. CK, respectively. Seven MYBs (CSS0030465, CSS0031744, CSS0000442, CSS0002706, CSS0006502, CSS0043569, and CSS0015964) were down-regulated in Cd-treated tea roots. Moreover, two MYBs (CSS0030465 and CSS0032956) and three AP2 (CSS0034513, CSS0009309, and CSS0043971) were down-regulated in As-treated tea roots. In addition, 45 DEGs with the same expression patterns, including 37 downregulated and eight upregulated DEGs, were identified in four pairwise comparison groups. Among seven upregulated DEGs, one ERF (CSS0000647) and six structural genes (CSS0033791, CSS0050491, CSS0001107, CSS0019367, CSS0006162, and CSS0035212) were significantly increased at 15 d of Cd and As treatments. WGCNA analysis revealed that the *CsERF*(CSS0000647) was positively correlated with five functional genes such as three *CsGolS*genes (CSS0001107, CSS0019367, and CSS0035212), one *CsNCED* (CSS0033791), and one *CsHIPP* (CSS0006162). One *CsCBL* (CSS0004428) was up-regulated in both Cd and As treatments, indicating that this gene may also play important roles in *C. sinensis* under the Cd and As treatments. These results help us to understand the mechanisms of Cd and As stresses response in tea plants and screen out the key candidate genes for future molecular breeding.

## Figures and Tables

**Figure 1 plants-12-01182-f001:**
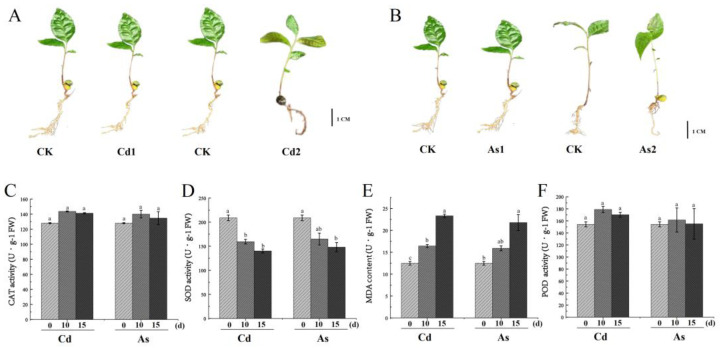
The effect of heavy metal Cd and As on phenotype, antioxidant enzyme activities, and MDA content. (**A**) The phenotypic difference in roots between Cd treated-tea plants and the control group; (**B**) The phenotypic difference in roots between As treated-tea plants and the control group. (**C**–**F**) Antioxidant enzyme activity and MDA content in tea roots between Cd/As treated-tea plants and the control group; (**C**) CAT activity; (**D**) SOD activity; (**E**) POD activity; (**F**) MDA content. Three biological replicates were obtained for each data point. Data were presented as means ± Sd (*N* = 3). Different letters above bars indicate significant differences between the Cd-treated *C. sinensis* roots and the control group (*p* < 0.05).

**Figure 2 plants-12-01182-f002:**
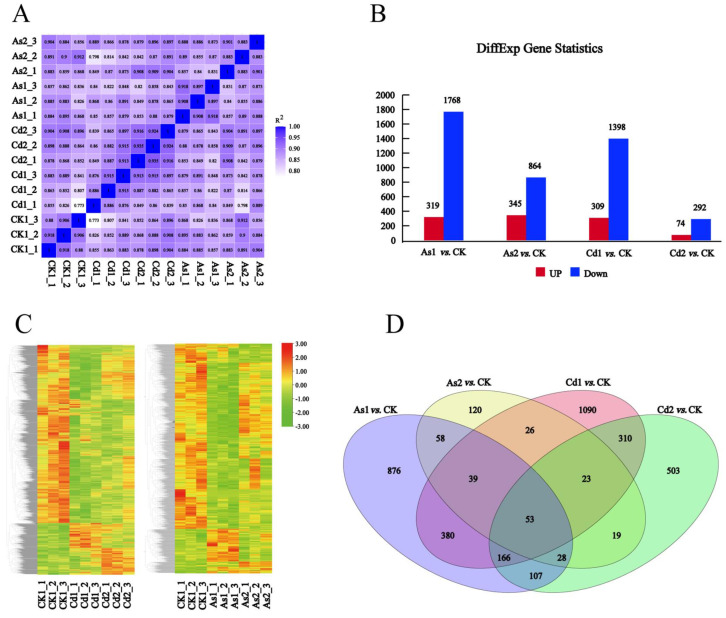
RNA-seq data analysis. (**A**) The correlation heat map of samples. The left gradient barcode color indicates the minimum value (0) in lavender and the maximum (1) in light blue. A value close to 1 indicates a high positive correlation, while a value close to 0 means no correlation. The heat map was drawn with DEseq (v3.5.1); (**B**) The number of up- or down-regulated DEGs in different pairwise comparisons. Blue and red columns represent the down- and up-regulated DEGs, respectively; (**C**) The expression patterns of DEGs between Cd-treated tea roots and the control group (right) and between As-treated tea roots and the control group; (**D**) Venn diagram of the DEGs among different pairwise comparisons. CK, control group; Cd1 and Cd2 indicate the tea roots treated with Cd for 10 and 15 days, respectively; As1 and As2 indicate the tea roots treated with As for 10 and 15 days, respectively.

**Figure 3 plants-12-01182-f003:**
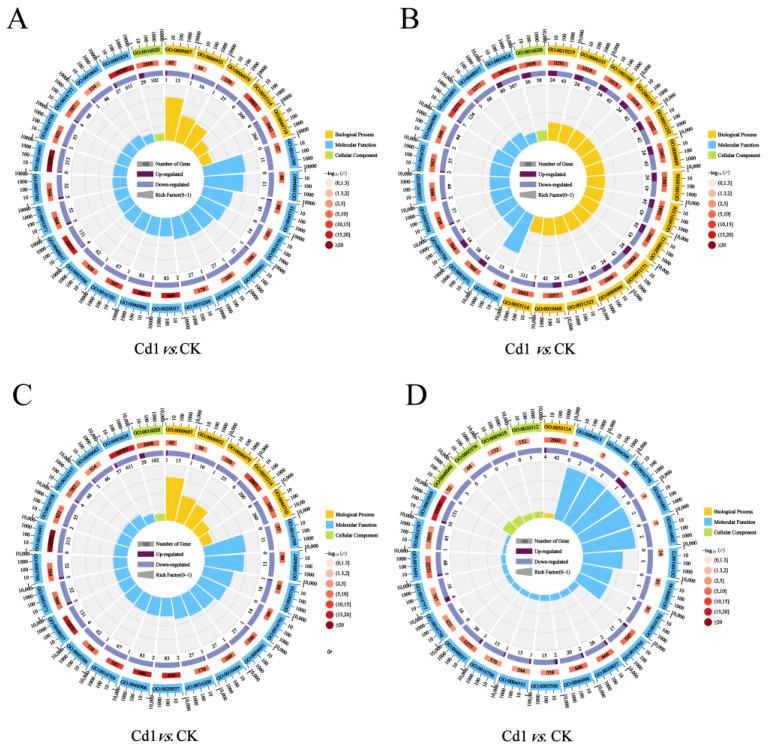
Identification and GO enrichment of DEGs. (**A**–**D**) Top 25 GO enrichment results of DEGs in different pairwise comparisons. (**A**) Cd1 vs. CK; (**B**) Cd2 vs. CK; (**C**) As1 vs. CK; (**D**) As2 vs. CK. CK, control group; Cd1 and Cd2 indicate the tea roots treated with Cd for 10 and 15 days, respectively; As1 and As2 indicate the tea roots treated with As for 10 and 15 days, respectively.

**Figure 4 plants-12-01182-f004:**
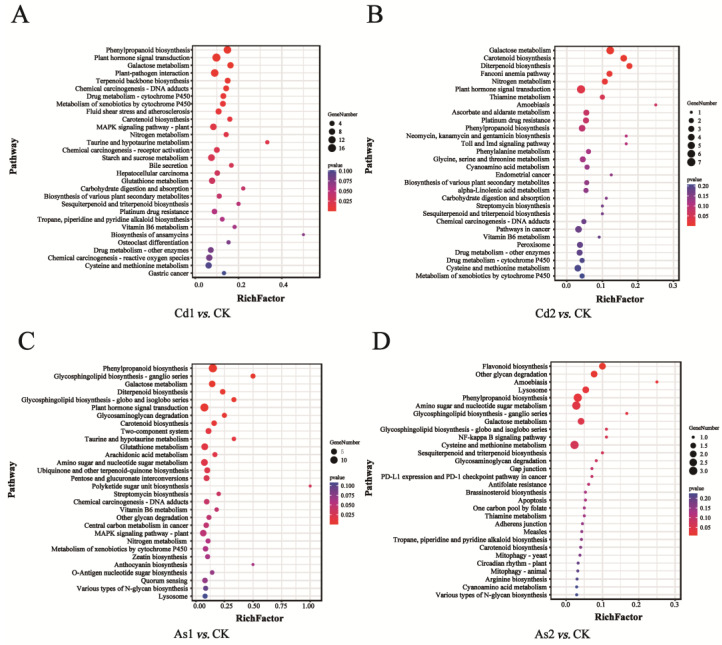
Identification and KEGG enrichment of DEGs. (**A**–**D**) Top 25 KEGG enrichment results of DEGs in four different pairwise comparisons. (**A**) Cd1 vs. CK; (**B**) Cd2 vs. CK; (**C**) As1 vs. CK; (**D**) As2 vs. CK. CK, control group; Cd1 and Cd2 indicate the tea roots treated with Cd for 10 and 15 days, respectively; As1 and As2 indicate the tea roots treated with As for 10 and 15 days, respectively.

**Figure 5 plants-12-01182-f005:**
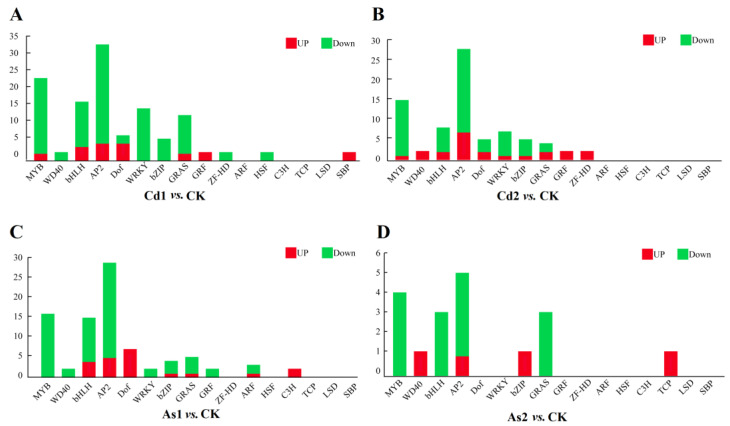
Differentially expressed TFs in Cd/As-treated tea roots. (**A**–**D**) Differentially expressed TFs in four different pairwise comparisons. (**A**) Cd1 vs. CK (**B**) Cd2 vs. CK; (**C**) As1 vs. CK; (**D**) As2 vs. CK. CK, control group; Cd1 and Cd2 indicate the tea roots treated with Cd for 10 and 15 days, respectively; As1 and As2 indicate the tea roots treated with As for 10 and 15 days, respectively. The red and green rectangles represented the upregulated and downregulated transcription factors, respectively.

**Figure 6 plants-12-01182-f006:**
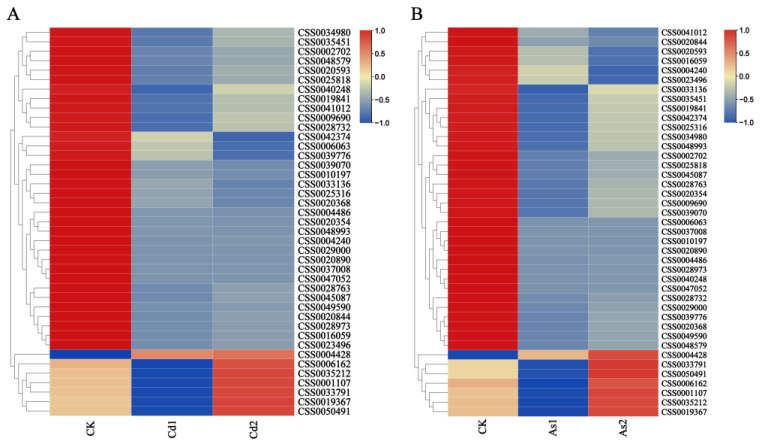
Expression patterns of DEGs in tea roots under Cd and As stresses. (**A**) Expression patterns of DEGs in tea roots under Cd stresses; (**B**) Expression patterns of DEGs in tea roots under As stresses. The scale bars represent the log_2_ transformations of the FPKM values. The colors from red to blue indicated the highest to lowest log_2_ (FPKM) values of each gene under the heavy metals Cd and As treatments.

**Figure 7 plants-12-01182-f007:**
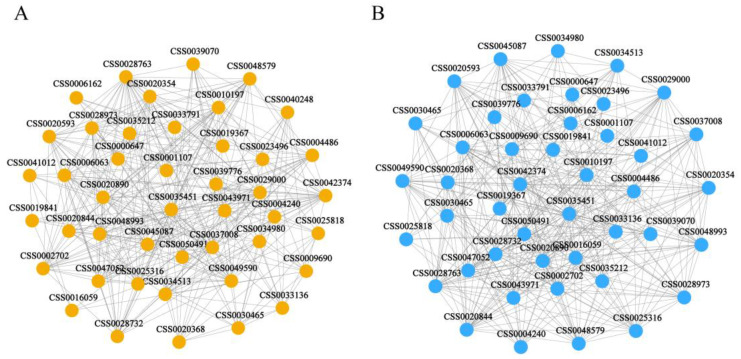
Co-expression analysis of DEGs identified in both Cd/As-treated tea roots. (**A**) Co-expression analysis of DEGs identified in Cd-treated tea roots;(**B**) Co-expression analysis of DEGs identified in As-treated tea roots. Nodes represent ‘genes’ and edges represent ‘relationships’ between any two genes. It was determined by a Pearson correlation coefficient > 0.80 or a Pearson correlation coefficient <−0.80, respectively.

**Figure 8 plants-12-01182-f008:**
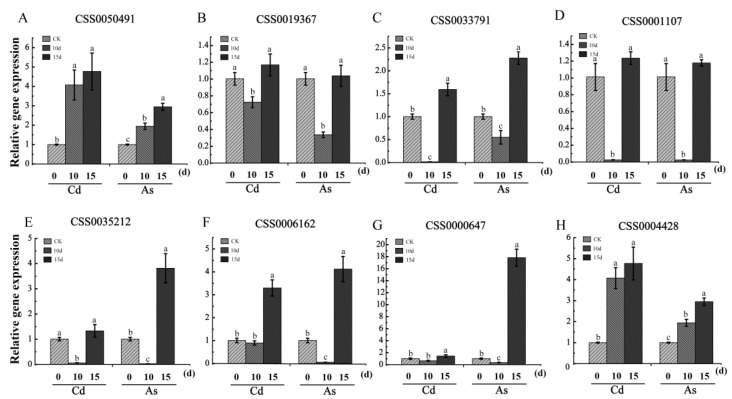
Validation of the expression patterns of eight candidate DEGs in Cd/As-treated tea roots. (**A**–**H**) The relative expression levels of candidate genes relative to *CsGAPDH* was measured by qRT-PCR. (**A**) The relative expression levels of CSS0050491 under Cd/As stress; (**B**) The relative expression levels of CSS0019367 under Cd/As stress; (**C**) The relative expression levels of CSS0033791 under Cd/As stress; (**D**) The relative expression levels of CSS0001107 under Cd/As stress; (**E**) The relative expression levels of CSS0035212 under Cd/As stress; (**F**) The relative expression levels of CSS0006162 under Cd/As stress; (**G**) The relative expression levels of CSS0000647 under Cd/As stress; (**H**) The relative expression levels of CSS0004428 under Cd/As stress. Three technical and biological replicates were used for each data point. Data were presented as means ± Sd (*N* = 3). Different letters above bars indicate significant differences between the Cd-treated tea roots and the control group (*p* < 0.05).

**Table 1 plants-12-01182-t001:** Summary statistics of RNA-seq results in Cd/As-treated tea roots.

Sample	Raw_Reads	Clean_Reads	Total_Map	Unique_Map	Clean_Bases	Expressed Gene	Standard Deviation	Q20	Q30	GC
CK1_1	44544050	43441036	36196019 (83.32%)	30927246 (71.19%)	6.52G	39153	295.250181	96.67	91.27	44.83
CK1_2	39422132	38291090	30179804 (78.82%)	25585746 (66.82%)	5.74G	38538	288.251178	96.66	91.24	45.24
CK1_3	44068264	43036462	34864588 (81.01%)	29517045 (68.59%)	6.46G	39869	295.151316	96.61	91.15	44.65
Cd1_1	47235258	46591286	37858289 (81.26%)	32536700 (69.83%)	6.99G	36481	320.939766	96.45	90.72	44.36
Cd1_2	41803762	40917046	33155043 (81.03%)	28292613 (69.15%)	6.14G	39746	298.284826	96.63	91.1	44.37
Cd1_3	46491206	45474714	37656708 (82.81%)	32106147 (70.6%)	6.82G	39366	292.976277	96.59	91.06	44.15
Cd2_1	45061350	44196684	37037737 (83.8%)	31537391 (71.36%)	6.63G	39580	293.200841	96.71	91.3	44.18
Cd2_2	39387626	38261120	30973412 (80.95%)	26334446 (68.83%)	5.74G	39212	290.471519	96.68	91.25	44.51
Cd2_3	44885852	43920534	35142943 (80.01%)	29969916 (68.24%)	6.59G	38744	308.138371	96.41	90.72	44.98
As1_1	42884692	42243424	33337391 (78.92%)	28307563 (67.01%)	6.34G	37658	294.061573	96.8	91.41	45.16
As1_2	42461254	41791448	35005786 (83.76%)	29926631 (71.61%)	6.27G	35675	297.560596	96.45	90.75	44.8
As1_3	46710062	45841500	37211971 (81.18%)	31459590 (68.63%)	6.88G	37408	292.188884	96.67	91.13	44.82
As2_1	41501826	40452076	30506971 (75.42%)	25955508 (64.16%)	6.07G	39304	293.338423	96.72	91.34	45.27
As2_2	42535242	41538986	34033550 (81.93%)	28852578 (69.46%)	6.23G	38725	292.120116	96.62	91.14	44.63
As2_3	40103436	38715324	31870755 (82.32%)	27049031 (69.87%)	5.81G	59129	296.849423	96.77	91.45	45.17

## Data Availability

The data presented in the study are deposited in the NCBI GEO repository, accession number GSE212827. Functional annotation of eight candidate genes in *C. sinensis*: Appendix A. Protein sequence of eight candidate genes in *C. sinensis*: Appendix A.

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
