# Peer review of "Transcriptome Analysis Reveals Differentially Expressed Genes Involved in Cadmium and Arsenic Accumulation in Tea Plant (Camellia sinensis)"

_plants, 2023, doi:10.3390/plants12051182_

Round 1

Reviewer 1 Report

Overall, this manuscript read well. However, there was not much new information on the metal stress response mechanism because broad spectra of various genes were suggested in this research, as shown in other publications. A model regarding ERF’s role related to those five functional genes will be very helpful to highlight your findings since there was no functional analysis of the ERF (CSS0000647). 

1.       Line 15; spell out the scientific name when it appears for the first time

2.       Line 30; show the experimental evidence to claim it

Introduction

3.       What is your hypothesis for this research?

4.       Provide any common or different genes found in your results compared to other experiments, including Tea (citation 24)

5.       Lines 109 and 115; content or activity?

6.       Figures 1 and 8; relabel mean separation letters

7.       Overall, figures and legends are not legible, too small to read

8.       Lines 317 – 318; rewrite the sentence

9.       Lines 345 – 347; elaborate further on how structural genes can play regulatory roles.

10.   How many replications were performed? Was there any statistical difference between/among replications?

Author Response

Reviewer1

I did not see the authors' response to my main question, although the authors revised many. In the abstract, "ERF might play important roles in regulating the heavy metals Cd and As stresses response by increasing the expression levels of GloS, NCED, and HIPP genes". Is an increase in transcription level directly related to regulation? There are many ways of regulations. This research looked at only the level of transcripts. At least the authors show/discuss these genes (ERF, GloS, NCED, and HIPP in terms of responding to As and Cd stress) are regulated at the transcriptional level. Also, I was expected to see a discussion (or a model) on how ERF could play a role in As and Cd stress response in relation to GloS, NCED, and HIPP genes.

My suggestion is the goal of this research was 'to identify candidate genes for crop development', instead.

Response1: Thank you for your comments. We are sorry for our misunderstanding of your main question. As the reviewer said, our study looked at only the level of transcripts, and we can not conclude that "ERF might play important roles in regulating the heavy metals Cd and As stresses response by increasing the expression levels of GloS, NCED, and HIPP genes". In our study, the results of WGCNA analysis revealed that the ERF transcription factor (CSS0000647) was positively correlated with five functional genes such as three GolS genes (CSS0001107, CSS0019367, and CSS0035212), one NCED (CSS0033791), and one HIPP (CSS0006162), which suggested that ERF transcription factor might also play important roles in enhancing the tolerance to Cd and As stresses. Through results of WGCNA analysis, we can not conclude that ERF could play a role in As and Cd stress response in relation to GloS, NCED, and HIPP genes. We have replaced these content with “the results of WGCNA analysis revealed that the ERF transcription factor (CSS0000647) was positively correlated with five functional genes such as three GolS genes (CSS0001107, CSS0019367, and CSS0035212), one NCED (CSS0033791), and one HIPP (CSS0006162), which suggested that ERF transcription factor might also play important roles in enhancing the tolerance to Cd and As stresses” in our manuscript.

Response2: We also agreed with the reviewer’s opinion that the goal of our research was 'to identify candidate genes for crop development'. We have modified our manuscript according to the reviewer’s suggestion, and hope that the corrections and response answered the reviewer’s main question.

We tried our best to improve the manuscript and made changes in our manuscript. We appreciate for the reviewers’ warm work earnestly, and hope that the corrections will meet with the approval of this journal. If you have any queries, please don’t hesitate to contact me at the address below.

Thank you and best regards.

Yours sincerely,

Xiaojing Wang

---------------

Dr. Xiaojing Wang

Professor

College of Tea Science,

Guizhou University,

Guiyang 550025, Guizhou Province, China.

Email: xjwang8@gzu.edu.cn

Reviewer 2 Report

The ms. contains many terms which are technical abbreviations which are not necessarily easy to understand for non-experts.

1.Especially some that appear in the abstract should be better defined there. For example, without reading the ms. further it is unclear from the abstract what Cd1, CK etc. mean. Same for ERF (at least it should be mentioned that it is a transcription factor). In addition, the sentence beginning line 20 is unclear: I guess the beginning of the sentence means only increased after 15 d (but not after 10 d), while the end of the sentence means in both Cd and As treatments after 10 and 15 d? Line 25, what exactly does ‘functional gene’ mean?

2. Similarly, line174, MF should be defined as ‘molecular function’: ‘molecular function (MF)’.

3.Line 264, ‘is localized’, not ‘are’.

4.I find that in the present version, many figures are far too small to be analyzed (fig. 2, 3, 4 and 7).

5.Concerning the data themselves, how do the authors explain the weak gene expression of many genes at 10 d in the RT-PCR data, and how does this correlate, or not, with the transcriptomic data.

6.Before publication can be considered, the public accessibility of the gene sequences must be demonstrated. When I type the gene names (such as CSS0000647) on Google I do not spontaneously find a database with the sequence. The authors should specify the pass to access these sequences.

Author Response

Dear Prof. Reviewer 2,

On behalf of my co-authors, we thank you very much for giving us an opportunity to revise our manuscript. We have improved our manuscript based on the comments you and reviewers provided. We appreciate for your warm work earnestly, and hope that the corrections will meet with approval. The responds to the comments are as follows:

Reviewer 2

Comments and Suggestions for Authors

The ms. contains many terms which are technical abbreviations which are not necessarily easy to understand for non-experts.

Response: Thank you for your suggestions and comments. We are sorry that we did not describe the full name of some terms, in the revised manuscript,We have added full names of some terms in the manuscript, please see.

  1. Especially some that appear in the abstract should be better defined there. For example, without reading the ms. further it is unclear from the abstract what Cd1, CK etc. mean. Same for ERF (at least it should be mentioned that it is a transcription factor). In addition, the sentence beginning line 20 is unclear: I guess the beginning of the sentence means only increased after 15 d (but not after 10 d), while the end of the sentence means in both Cd and As treatments after 10 and 15 d? Line 25, what exactly does ‘functional gene’ mean?

Response: Thank you for your suggestions and comments. We have added the full names of the scientific name in the abstract, and further rewritten the abstract, please see Lines 11-30.

Lines 11-30: Abstract: Tea (Camellia sinensis) is the second most consumed drink in the world. Rapid industrialization has caused various impacts on nature and increased pollution by heavy metals. However, the molecular mechanisms of cadmium (Cd) and arsenic (As) tolerance and accumulation in tea plants are poorly understood. The present study focused on the effects of heavy metals Cd and As on tea plants. Transcriptomic regulation of tea roots after Cd and As exposure was analyzed to explore the candidate genes involved in Cd and As tolerance and accumulation. In total, 2087, 1029, 1707, and 366 differentially expressed genes (DEGs) were obtained in Cd1 (with Cd treatment for 10 days) vs. CK (without Cd treatment), Cd2 (with Cd treatment for 15 days) vs. CK, As1 (with As treatment for 10 days) vs. CK (without Cd treatment), and As2 (with As treatment for 15 days) vs. CK, respectively. Analysis of DEGs showed that a total of 45 DEGs with the same expression patterns were identified in four pairwise comparison groups. One ERF transcription factor (CSS0000647) and six structural genes (CSS0033791, CSS0050491, CSS0001107, CSS0019367, CSS0006162, and CSS0035212) were only increased at 15 d of Cd and As treatments. Using weighted gene co-expression network analysis (WGCNA) revealed that the transcription factor (CSS0000647) was positively correlated with five structural genes (CSS0001107, CSS0019367, CSS0006162, CSS0033791, and CSS0001107). Moreover, one gene (CSS0004428) was significantly upregulated in both Cd and As treatments, suggesting that these gene played a positive regulatory role in enhancing Cd and As tolerance. These results provide a theoretical basis for the use of genetic engineering technology to modify plants by enhancing multi-metal tolerance to promote phytoremediation efficiency.

  1. Similarly, line174, MF should be defined as ‘molecular function’: ‘molecular function (MF)’.

Response: Thank you for your valuable comments. We are sorry for the confusion caused by this parts, we have changed ‘MF’ into ‘molecular function (MF)’ in the revised manuscript, please see lines 166-168:

Lines 166-168: Our results showed that the top 25 enriched GO terms of the DEGs in Cd1 vs. CK were divided into three GO categories, including 5 biological process categories,1 cellular component category, and 19 molecular function (MF) categories.

  1. Line 264, ‘is localized’, not ‘are’.

Response: Thank you very much for your valuable comments. We have changed ‘are’ into localized in the revised manuscript, please see lines 256-257.

Lines 265-266: The ERF transcription factor (CSS0000647) is localized at the center of the network.

  1. I find that in the present version, many figures are far too small to be analyzed (fig. 2, 3, 4 and 7).

Response: Thank you for your comments. In the revised manuscript, we have revised all figures and legends in the manuscript. Please see Figure 2, Figure 3, Figure 4 and Figure 7.

Figure 2 RNA-seq data analysis. (A) The correlation heat map of samples. The left gradient barcode color indicates the minimum value (0) in lavender and the maximum (1) in light blue. A value close to 1 indicates a high positive correlation, while a value close to 0 means no correlation. The heat map was drawn with DEseq (v3.5.1). (B) The number of up- or down-regulated DEGs in different pairwise comparisons. Blue and red columns represent the down- and up-regulated DEGs, respectively. (C) The expression patterns of DEGs between Cd-treated tea roots and the control group (right) and between As-treated tea roots and the control group. (D) Venn diagram of the DEGs among different pairwise comparisons. CK, control group; Cd1 and Cd2 indicate the tea roots treated with Cd for 10 and 15 days, respectively; As1 and As2 indicate the tea roots treated with As for 10 and 15 days, respectively.

Figure 3 Identification and GO enrichment of DEGs. (A-D) Top 25 GO enrichment results of DEGs in different pairwise comparisons. A: Cd1 vs. CK; B: Cd2 vs. CK; C: As1 vs. CK; D: As2 vs. CK. CK, control group; Cd1 and Cd2 indicate the tea roots treated with Cd for 10 and 15 days, respectively; As1 and As2 indicate the tea roots treated with As for 10 and 15 days, respectively.

Figure 4 Identification and KEGG enrichment of DEGs. (A-D) Top 25 KEGG enrichment results of DEGs in four different pairwise comparisons. A: Cd1 vs. CK; B: Cd2 vs. CK; C: As1 vs. CK; D: As2 vs. CK. CK, control group; Cd1 and Cd2 indicate the tea roots treated with Cd for 10 and 15 days, respectively; As1 and As2 indicate the tea roots treated with As for 10 and 15 days, respectively.

Figure 7 Co-expression analysis of DEGs identified in both Cd and As-treated tea roots. (A) Co-expression analysis of DEGs identified in Cd-treated tea roots. (B) Co-expression analysis of DEGs identified in As-treated tea roots. Nodes represent ‘genes’ and edges represent ‘relationships’ between any two genes. It was determined by a Pearson correlation coefficient > 0.80 or a Pearson correlation coefficient < -0.80, respectively.

  1. Concerning the data themselves, how do the authors explain the weak gene expression of many genes at 10 d in the RT-PCR data, and how does this correlate, or not, with the transcriptomic data.

Response: Thank you for your comments and suggestions. Our results showed that the expression levels of many genes were decreased at 10 d in the RT-PCR data, which was consistent with the results of transcriptome analysis results. The C. sinensis plants had strong heavy metal Cd and As resistance. Under mild stress, the expression levels of DEGs involved in the biosynthesis of various metabolites were decreased due to the strong heavy metal resistance of tea plants. Under moderate and severe Cd and As stress, the expression levels of DEG was the most affected.

  1. Before publication can be considered, the public accessibility of the gene sequences must be demonstrated. When I type the gene names (such as CSS0000647) on Google I do not spontaneously find a database with the sequence. The authors should specify the pass to access these sequences.

Response: Thank you for your comments and suggestions. In the revised manuscript, we have added the gene sequences in Table S7.

C. sinensis

A. thaliana

E-value

Function description

Source

Gene locus

Gene name

Gene locus

Gene name

CSS0050491

CsRS8

AT5G40390

AtRS5

0

Involved in response to drought and cold stress

doi: 10.3390/ijms24010202

CSS0019367

CsGolS

AT2G47180

AtGolS1

5.6E-149

Involved in detoxification of heavy metal cadmium

doi:10.1016/j.envexpbot.2023.105217

CSS0033791

CsNCED

AT3G14440

AtNCED3

4.20E-251

Involved in detoxification of heavy metal cadmium

doi.org/3389.2020/fpls.00909

CSS0001107

CsGolS

AT2G47180

AtGolS1

1.10E-147

Involved in detoxification of heavy metal cadmium

doi:10.1016/j.envexpbot.2023.105217

CSS0035212

CsGolS

AT2G47180

AtGolS1

5.70E-149

Involved in detoxification of heavy metal cadmium

doi:10.1016/j.envexpbot.2023.105217

CSS0006162

CsHMP

AT4G08570

AtHMP26

2.60E-48

Heavy metal transport/detoxification superfamily protein

-

CSS0004428

CsCBL7

AT5G55990

AtCBL2/3

1.10E-114

Involved in protecting plants from high Mg2+

doi: 10.1073/pnas.1420944112

CSS0000647

CsERF54

AT2G20880

AtERF53

7.90E-53

Involved in response to drought stress

doi 10.1007/s11103-013-0054-z

Supplementary Table 7 | Functional annotation of eight candidate genes in C. sinensis.

Supplementary Table 10 | Coding equences of eight candidate genes in C. sinensis

>CSS0000647 PREDICTED: Camellia sinensis ethylene-responsive transcription factor ERF054-like

ATGGATCAATCTGGGAATAGTGGCAAATCAAAAAAAGTTATTGAAGAAACTGAGAAGATGGCCCAGATGGAATGGGTGAGAGACCAAACTAAGGACTTCGACATCGGCTTCGAGAGGCCACAATGGAAGCCAGTTTTTGAAGAAGCTTCCATGTCAAGTAGGCCTCTCAAGAAGGTCAAAAGTCCTGAACGCCAAGACCCATTTCAATCCTCTGTTTCTTCAGCTCAACAAACTCCTTCTCTGTCTCTCTCATCCTCTGTTCCTTACACTACTACAGCACCCATAGCTTTCCCTCCATCATCATCTTCAAGGCTTGTATTTCCTTTTGCCTTGGATAATTCTCAACAATCCATGGAAAATCTACACCAATTCCAAACCAACCCGCCTCCTCTGTTCCTTCAACCGCCACAAAATCAACAGCAAATGATTTCCTTTGCTCCTCAGAACCAAAGCCTAGGCTTCCCACCATACTTTGCCGGAGACTCGGCTCCATCACAGCAGCAATTACTGCAGTACTGGAGTGATGCACTGAATTTGAGTCCAAGAGGAAGGATGATGATGATGAACAGGTTGGGACAGGATGGTCGGGGCTTGTTTAGGCCTCCAGTGCCCCCTATTTCACCTACCAAGCTCTACAGAGGAGTGAGGCAGAGACATTGGGGCAAATGGGTAGCTGAAATTCGTCTCCCTCGAAACAGGACTCGCCTCTGGCTCGGCACCTTTGACACAGCAGAGGACGCCGCCTTAGCCTATGATCGCGAGGCCTTTAAGTTGAGGGGAGAGAATGCACGGCTCAATTTCCCCGAACTCTTCCTCAATAAAGACAGAGCAGCTTCGTCAGCGCCCGGTTCATCTTCATCGTCTCCTCCTACTCCCCATGGAAATTTGGCTCCTAGCCGGTACTCGAAACAACCTCAGGAAGCTTCAGAAGGCCCAAATTTGGAGGTTTCAAGTATGGAAATGCCACCACCAGAGTGGCCCAAACCTCAAGAAGATAACCCAGATGAGGATTCGGGATTGGGGTCGAGCGAGGTCACAGTTGCAGGGGGTTCTGGAGCAGGGGAAGGTATTTTGGAGTCATCGGAATTGGTTTGGGGTGACATGGCAGAGGCTTGGATGAATGCAATTCCAGCTGGTTGGGGTCCAGGTAGTCCAGTTTGGGATGATTTGGACACAACCAATAATCTTCTGTTGCCTTCAAATCTTCCTTTTGCCAATGTTCATCAGCAGGACTTGAGTGGTTCTAATCCTCAAAAACAGGAACAGAACATCGCCCCGGCTGATTCATCGTCTTCTTCTTCTTCATGCCCTATGAAGCCCTTCTTTTGGAAGGATCAGGACTGA

>CSS0033791 PREDICTED: Camellia sinensis 9-cis-epoxycarotenoid dioxygenase NCED1

ATGGCTTCTTCAACAACACCTTCTACAAGTACATGGGTTAAGCCCAAATTGATCCCTACTTCCTCGAGTGCATTGGGCTCTTGTTCCACCTCCTCATTCTCTCTAAGCAAGCCCAATCCCAATAGGCAACAAAACATTGCTTGCTCTCTCCACACCCCTTCCATACGCCACCTTCCAAAGCACGCTCCCCCAACATATCACACACTCAGCACTGGTATTGCCACTAAAGATAACTCTTCCTTGTCCTCCACCTCTGCCAATGCTCAATCATGGAATTTCTTACAGAAAGCAGCGGCAATGGCGTTGGATGTAGCGGAGAGTATCCTAGTGGCACGCGAAAAAGAGCATCGTCTACCCAAGACGGCCGACCCGGGAGTCCAAATTTCCGGCAACTTCGCCCCGGTGCCGGAGCACTCGGTCCAGCACAACCTCCGCATCGCCGGAAAAATCCCCGAATGCATTCAAGGCGTTTACATCCGGAACGGAGCGAACCCGCTATTCGAGCCGGTTGCCGGCCACCACTTCTTCGACGGCGACGGCATGGTCCACGCCGTCAAATTCAATGACGGCGCGGCTAGCTACTCCTGCCGGTTCACCGAAACCCAGAGGCTCGTTCAGGAACGGGCCTTGGGCCGGCCCGTCTTCCCCAAAGCCATCGGCGAACTCCACGGCCACTCCGGCATCGCCAGGCTATTACTGTTCTACGCTCGCGGGATATTCGGGCTCGTCGATCACAGCCACGGCACTGGCGTAGCCAACGCCGGACTAGTCTACTTCAACAACCGATTACTCGCCATGTCGGAAGACGATTTGCCGTACCACGTCAGAGTCACCCCCTCCGGCGATCTCAGAACTGTGGGTAGGTATGATTTCGATAATCAACTCAATTCCACCATGATCGCTCATCCCAAACTCGACCCAGTTTCCGGCCAACTCTTCGCTCTCAGCTACGATGTGATCCAGAAACCATACCTCAAATACTTCAAATTTTCACCCTCCGGCGAGAAATCACCGGACGTTGAAATCCCACTCTCAGAACCAACCATGTTGCATGATTTCGCAATCACCGAGAAATTCGTAGTGATTCCTGATCAACAAGTAGTTTTCAAGCTCCCGGAGATGATCCGCGGTGGCTCGCCGGTGGTTTACGACAAAAACAAGATGTCGAGGTTTGGGATTTTGAACAAGAATGCCACTGATGCTTCTGAGATCAAATGGGTCGAAGCACCAGATTGCTTTTGCTTCCACCTCTGGAATGCTTGGGAAGAGCCAGAGACCGATGAGATCGTCGTGATTGGGTCTTGTATGACTCCACCGGACTCAATTTTCAACGAATGTGACGAGGGACTAAAGAGTGTTCTATCAGAAATCAGGTTCAATTTAAAGACTGGTAAGTCTTCTCGCAGACCCATAATGAGTGAGACTGATCAAGTGAATTTAGAGGCAGGAATGGTGAACCGGAACTATCTCGGCCGCAAATCCCAGTTCGCTTACCTCGCCATCGCCGAGCCGTGGCCTAAAGTCTCCGGATTCGCCAAAGTCGACCTCTCCACTGGCGAGGTGAAGAAACACAATTACGGCAATGAAAAATACGGTGGCGAGCCTCTGTTCCTGCCCAGAGACCCAAATTCAGTATCTGAAGATGATGGGTATATTCTAGCTTTTGTACACGACGAGAAGAAGTGGAAATCAGAGCTACAGATCGTGAACGCCATGAATTTAGAATTAGAAGCTACAATCGAGCTACCATCTCGAGTTCCCTATGGCTTTCATGGCACATTCATAAGTGCCAAAGACTTAGCCAAACAAGCCTAA

>CSS0050491 PREDICTED: Camellia sinensis probable galactinol-sucrose galactosyltransferase 5

ATGGCTCCAAGTTTGAGCAAAGCTGGCTCCGATATTTCAGACCTCGTCAATGGCTACACCGGCTCATCATCACCAACAATCACATTAGAGGGATCAAACTTCAAAGCCAGAGGCCACGTCATTCTCTCCGATGTACCTCCCAACATCGTAGCTTTTCCCTCACCCTACGGCGGCGGCGCCACCACCACCGCCGCCGGCAGCTTCGTCGGATTCGACTCCCGCGAGGCCAAAAGCCGCCACGTGGTGTCGGTGGGCAAGCTCAAGGGGATCACATTCATGAGTATTTTCAGATTCAAGGTCTGGTGGACGACCCACTGGGTCGGATCCAACGGCCGAGACCTCGAATCCGAAACCCAGATGGTGATTCTGGACCGCTCAGCCGAGGGCCGCCCTTACGTCCTTCTCCTCCCCCTCCTTGAAGGCCCATTCCGCGCCTCTCTCCAACCCTGCGGGTCGACAAAAGTGACCCGAGCTTCGTTTCGGAGCGTGCTTTACATTCACGCAAGCGACGATCCGTACAGTCTCGTGGAGGACGCCATGGAAGTAGTGAGAGCTCATCTCGGAACATTCAGGCTTTTGAAAGAGAAGACACCACCGGGTATCGTGGACAAATTTGGTTGGTGCACGTGGGATGCATTTTACCTTACGGTGCAACCTCAGGGAGTTTGGGAAGGTGTGAAAGGGCTCGTAGAAGGCGGGTGCCCACCGGGTCTGGTTCTTATTGATGATGGGTGGCAGTCCATATCCCATGATGATGACCCGATTGCCCAAGAAGGCATGAATCGGACCTCTGCTGGTGAGCAAATGCCGTGTAGGCTCATAAAGTTTGAAGAGAATTACAAATTTAGGGACTATGTGAGTCGTAAAAAGTCAACCTCTGGTGGTGCCCTTAATAAGGGCATGGGTGCCTTTATTAGGGACCTCAAGGGGGAGTTTAAGAGTGTGGATTATGTGTATGTGTGGCATGCTTTGTGTGGGTATTGGGGCGGGCTTAGACCCAATGCTTCGGGTCTACCCAAATCTAGAGTTGTGAGGCCTAAGTTGTCACCTGGGTTGGAGAAGACAATGGAAGATCTAGCCGTTGATAAGATCGTGAACAATGGAGTTGGGTTGGTCCCACCAGAGATGGTTGATCAAATGTATGAAGGTCTGCACTCGCATTTGGAGTCAGTTGGGATTGATGGAGTCAAAGTGGATGTTATTCACTTGTTGGAGATGTTGTGTGAGGACTATGGAGGCAGAGTAGAACTAGCAAAAGCATACTACAAAGCATTGACATCTTCAGTGAGGAAGCACTTCAATGGCCATGGTGTCATCGCAAGCATGGAACACTGCAACGACTTCATGTTCCTCGGAACCGAAGCCATAGCTCTCGGTCGTGTCGGGGATGATTTTTGGTGCACTGATCCATCTGGGGATCCCAATGGGACCTTTTGGCTACAAGGGTGTCACATGGTGCACTGTGCTTATAACAGCTTGTGGATGGGGAACTTCATACACCCAGATTGGGACATGTTCCAATCCACTCATCCTTGTGCCCAGTTCCACGCCGCCTCTCGGGCTATTTCCGGTGGACCAATTTATGTCAGCGACTCTGTCGGAAAACACGATTTCAAACTGCTCAGGAGCCTGGTGTTACCTGATGGCTCGATCTTGCGGTGCGAGCACTATGCACTTCCTACCCGTGATTGTCTCTTCCAAGACCCTCTCCACGATGGGAAGACCATGCTCAAGATATGGAACTTAAACAAGTACACTGGAGTTCTTGGAGCATTTAACTGCCAAGGTGGAGGATGGTGTCGCGAGTCCAGGCGCAACAAATGCGCCTCCGACTGTTCCCACACCGTGACCACAACCACCACCCCAAAAGACATTGAGTGGAAACGTGGAACAAAGCCAATCTCCATTGAAGCAGTGCAACTATTTGCATTGTACATGTTCCGCGAGAAAAAATTAATTCTCTCAAAGCCATCCGACGCCATAGAGATCTCACTCGAACCCTTTAACTTTGAGCTTATAACGGTCTCTCCAATCACCACCATGGCTAACAAGTCTATCCAATTTGCTCCGATTGGGCTGGTGAATATGCTTAACACTGGCGGTGCAATACAGTCAGTGGTGTTCAATGATCGGGACAACTCGGTCCAGGTAGGGGTGAAGGGGACTGGTGAAATGAGGGTGTTTGCGTCGGAGAAACCGACAGCTTGCCGGCTTAATGGGGAAGATGTTGAATGTGGTTATGAGGGGCTCATGGTTGTAGTTCAAGTGCCATGGCCTGAATCTTCAAGTTTGTCTATTATTGAGTACTCATTTTAA

>CSS0001107 PREDICTED: Camellia sinensis galactinol synthase 2-like

ATGGCTCCTGAACTTGTGAGTGCAACCACCAAGGCCACCAGCCTAGTCAAGGCCGCGAGCCTGATGAGCAGAGCTTATGTCACTTTCTTGGCTAGCAATGGTGACTATGTCCAGGGCGTGGTCGGCCTAGCCATGGGCCTGAGGAAGGTCCACACTGCCTACCCGCTTGTGGTGGTCGTGCTTCCGGATGTGCCGGAGGAACACCGCCGTATGCTGGTGGCTCAGGGATGCATAACCCAGTTTGCTATGGCCTACTATGCCATCAACTACTCCAAGCTTCGCATCTGGGAGTTTGTGGAATACCATAAGATGATATACTTGGACGGAGACATCCAGGTGTTTGAAAACATAGATCACCTCTTTGACAATCCAAATGGCTACTTTTATGCTGTGAAGGACTGTTTTTGTGAAAAGACATGGTCTCACACCCCACAGTACCAGATTGGGTACTGCCAACAGTGTCCTGATAAGGTCCAGTGGCCTGCAGAGTTGGGTCCCAAGCCTCCCCTCTACTTCAATGCTGGCATGTTCATGTTCGAGCCCAGTCTCTCAATTTATGATGACCTCTTAGAGACCCTCAAAATCACCCCGCCTACCCCTTTTGCTGAGCAGAAAAACCCACAAACTCACCAAAGCATCCCCACATTTCTTCTTCATTTTCTCAAAAACACTTCCCAACTTCCTTTCTCTCTCTACAGACCTCTCTTTCTTTCTCTAAAACTTGTTGTGAAAACTCAAATGGCTCCTGAACTTGTGAGTGCAACCACCAAGGCTACCAGCCTGGTCAAGGCAGCCAGCCTGACGAGCCGGGCTTATGTGACATTCTTGGCAGGGAATGGTGACTACGTGAAGGGCGTGGTTGGGCTGGCCAAGGGCCTGAGGAAGGTCAAGACTGCGTACCCGCTAGTGGTGGCGGTCCTGCCTGACGTGCCTGAGGAGCACCGCCGTATGCTGTTAGCTCAGGGCTGCATAGTCCGTGAGATCGAGCTGGTGTACCCGCCAGCGAACCAGACCCAGTTTGCTATGGCCTACTATGTCATCAACTACTCCAAGCTTCGTATCTGGGAGTTTGTGGAGTACCATAAGATGATATATTTGGACGGAGACATCCAGGTGTTTGAAAATATAGACCACCTCTTTGACAACCCAAATGGCTACTTCTATGCTGTGAAGGATTGTTTCTGCGAGAAGACATGGTCTCATACCCCACAGTACCAGATTGGGTACTGCCAACAGTGCCCTGATAACGTCCAGTGGCCTAGAGAGTTGGGTCCTAAGCCTCCCCTCTACTTCAATGCTGGCATGTTTGTGTTCGAGCCTAGTCTCCCAATTTATGATGACCTCTTAGAGACCCTCAAAATCACTCCGCCTACCCCTTTTGCTGAGCAGGACTTTTTGAATATGTTCTTTAAGGATGTCTACAAGCCTATCCCACCAATTTACAACCTTGTTTTGGCCATGCTTTGGCGTCACCCAGAAAACATTGAACTTAATAAAGTGAAAGTTGTTCACTACTGTGCCGCGGGATCAAAGCCATGGAGGTATACTGGAATTGAAGAGAACATGAAAAGGAAAGACATTAAGATGCTAGTGAAGAGCTGGTGGGATATATACAGTGATGAGTCATTAGATTATAAGAAGACTAACACATCGCCCATTGAAGGTGAAGCTGACAAATTCATGGCGGCAACGGTGTCAGAGGCTAACGCTGGTCATTACATTACCGCCCCATCTGCTGCCTAG

>CSS0019367 PREDICTED: Camellia sinensis galactinol synthase 2-like

ATGGCTCCTGAACTTGTGAGTGCAACCACCAAGGCTACCAGCCTGGTCAAGGCAGCCAGCCTGACGAGCCGGGCTTATGTGACATTCTTGGCAGGGAATGGTGACTACGTGAAGGGCGTGGTTGGGCTGGCCAAGGGCCTAAGGAAAGTCAAGACTGCGTACCCGCTAGTGGTGGCGGTCCTGCCTGACGTGCCGGAGGACCACCGCCGTATGCTGCTAGCTCAGGGCTGCATAGTCCGTGAGATCGAGCCGGTGTACCCACCGGCGAACCAGACCCAGTTTGCTATGGCCTACTATGTCATCAACTACTCCAAGCTTCGTATCTGGGAGTTTGTGGAGTACCATAAGATGATATATTTGGACGGAGACATCCAGGTGTTTGAAAATATAGACCACCTCTTTGACAACCCAAATGGCTACTTCTATGCTGTGAAGGATTGTTTCTGCGAGAAGACATGGTCTCATACCCCACAGTACCAGATTGGGTACTGCCAACAGTGCCCTGATAAGGTCCAGTGGCCTAGAGAGTTGGGTCCTAAGCCTCCCCTCTACTTCAATGCTGGCATGTTTGTGTTCGAACCTAGTCTCCCAATTTATGATGACCTCTTAGAGACCCTCAAAATCACTCCGCCTACCCCTTTTGCTGAGCAGACCACCCTTTCTTTCTCTAAAACTCAAATGGCTCCTGAACTTGTGAGTGCAACCACCAAGGCTACCAGCCTGGTCAAGGCAGCCAGCCTGACGAGCCGGGCTTATGTGACATTCTTGGCAGGGAATGGTGACTACGTGAAGGGCGTGGTTGGGCTGGCCAAGGGCCTGAGGAAGGTCAAGACTGCGTACCCGCTAGTGGTGGCGGTCCTGCCTGACGTGCCGGAGGACCACCGCCGTATGCTGCTAGCTCAGGGCTGCATAGTCCGTGAGATCGAGCCGGTGTACCCACCGGCGAACCAGACCCAGTTTGCTATGGCCTACTATGTCATCAACTACTCCAAGCTTCGTATCTGGGAGTTTGTGGAGTACCACAAGATGATATATTTGGACGGAGACATCCAGGTGTTTGAAAATATAGACCACCTCTTTGACAACCCAAATGGCTACTTCTATGCTGTGAAGGATTGTTTCTGCGAGAAGACATGGTCTCATACCCCACAGTACCAGATTGGGTACTGCCAACAGTGCCCTGATAAGGTCCAGTGGCCTAGAGAGTTGGGTCCTAAGCCTCCCCTCTACTTCAATGCTGGCATGTTTGTGTTCGAGCCTAGTCTCCCAATTTATGATGACCTCTTAGAGACCCTCAAAATCACTCCGCCTACCCCTTTTGCTGAGCAGGACTTTTTGAATATGTTCTTTAAGGATGTCTACAAGCCTATCCCACCAACTTACAACCTTGTTTTGGCCATGCTTTGGCGTCACCCAGAAAACATTGAACTTAATAAAGTGAAAGTTGTTCACTACTGTGCCGCGGGATCAAAGCCATGGAGGTATACTGGAATTGAAGAGAACATGAAAAGGAAAGACATTAAGATGTTAGTGAAGAGCTGGTGGGACATATACAATGATGAGTCATTGGATTATAAGAAGACTAACACATCACCCATTGAAGGTGAAGCTGACAAATTCATGGCGGCAACGGTGTCAGAGGCTAACCCTGGTCATTACATTACCGCCCCATCTGCTGCCTAG

>CSS0006162 PREDICTED: Camellia sinensis heavy metal-associated isoprenylated plant protein 23-like

ATGGGAGTTTCAGGCACTTTGGAGTACTTCTCTGAATTGCTGAGCAGTGTCAAGAAGGTCAAGAAAATGAAACAGTTAAACACTGTGGCTCTCAAAGTCAGGATGGACTGTGAGGGTTGTAAACGTAAAGTCAAAAACACACTCTCTTCTCTCAAAGGAGTGAAATCAGTGGATGTGGATTTGAAGGAGCAGAAGGTGACAGTGAGTGGATATGTGGATGCAAAGAAAGTGCTGAAGAAGGCTCAATCTACTGGTAACAAATCTGAGCCATGGCCATATGTTCCATACAACCTGATAGCTCACCCATACGCTGCTCAAGTTTATGACAAGAAGGCACCTCGTGGTTTTGTTAGGGGCACTGAAGACACTGCCATTGCTACTCTCAGCACGGTTGAACAACACTACACCACCATGTTCAGTGATGATAATCCTAATGCTTGCTCTATCATGTAGC

>CSS0004428 PREDICTED: Camellia sinensis calcineurin B-like protein 3

ATGTTGCAGTGCTTAGAAGGGGTTAAGCATTTGTGTGCTTCCCTACTTCGCTGCTGTGATCTTGATTTGTATAGACAATCCAGAGACCTTGAAGATCCTGAAGTTCTTGCAAGGGAGACAGTGTTTAGTGTAAGTGAAATAGAGGCACTTTATGAGCTATTTAAAAAGATTAGCAGCGCAGTAATCGATGACGGGCTGATCAACAAGGAAGAGTTTCAACTGGCATTATTTAAGACAAACAAAAAGGAGAGCTTATTTGCTGATCGGGTATTTGACTTGTTTGACACAAAACATAATGGCATCCTAGGCTTTGAAGAGTTTGCTCGTGCACTCTCTGTATTTCATCCAAATGCCCCTATTGATGATAAGATTGAGTTTTCGTTCCAACTGTACGATCTCAAGCAGCAAGGTTTCATTGAAAGGCAAGAGGTGAAGCAAATGGTAGTGGCGACTCTTGCTGAATCTGGTATGAACCTTTCAGATGATGTTATAGAGAGTATCATCGACAAGACCTTTGAGGAAGCTGATACGAAACATGATGGGAAGATTGACAAGGAAGAGTGGAGAAGTCTTGTACTGCGACATCCATCCCTTTTGAAGAATATGACCCTCCAGTACCTCAAGGACATCACCACTACATTCCCAAGCTTCGTATTTCATTCACGGGTCGAGGATACCTGA

>CSS0035212 REDICTED: Camellia sinensis galactinol synthase 2-like

ATGTCTCCTGAACTTGTGAGTGCAACCACCAAGGCCACCAGCCTAGTCAAGGCCGCGAGCCTGATGAGCAGAGCTTATGTCACTTTCTTGGCTGGCAATGGTGACTATGTCAAGGGCGTGGTCGGGCTAGCCAAGGGCCTGAGGAAGGTCCACACTGCCTACCCGCTTGTCGTGGCCGTGCTTCCGGATGTTCCGGAGGAACACCGCCATATGCTGGTGGCTCAGGGATGCATAGTCCGTGAGATCGTGCCGGTGTACCCGCCGGAGAACCAGACCCAGTTTGCTATGGCCTACTATGTCATCAACTACTCCAAGCTTCGCATCTGGGAGTTTGTGGAGTACCATAAGATGATATACTTGGACGGAGACATCCAGGTGTTTGAAAACATAGATCACCTCTTTGACAATCCAAATGGCTACTTTTATGCCGTGAAGGACTGTTTTTGTGAAAAAACATGGTCTCACACCCCACAGTACCAGATTGGGTACTGCCAACAGTGCCCTGATAAGGTCCAGTGGCCTGCAGAGTTGGGTCCCCGGCCTCCCCTCTACTTCAATGCTGGCATGTTCGTGTTCGAGCCCAGTCTCTCAATTTATGATGACCTCTTAGAGACCCTCAAAATCACCCCGCCTACCCCTTTTGCTGAGCAGGACTTTTTGAATATGTTCTTTAGGGATGTTTACAAGCCTATCCCACCAATTTACAACCTTGTTTTGGCCATGCTTTGGCGCCACCCAGAGAACATTGAACTCAACAAAGTGAAAGTTGTTCACTATTGTGCCGCGGGATCAAAGCCATGGAGGTATACTGGAAAAGAAGAGAACATGGATAGAAAAGATATCAAGATGCTAGTGAAGAGTTGGTGGGACATATACAATGATGAATCATTGGACTACAAGAGGACTAACATATCACCCATTGAAGGTGAAGCCGACAAAGTCATAGTGACAATGGTGGCAGAGGCCAACGTTGGTCATTACACTCCCGCTCCACCTGCCGCCTAG

We tried our best to improve our manuscript, and hope that the corrections will meet with the approval of this journal. If you have any queries, please don’t hesitate to contact me at the address below.

Thank you and best regards.

Yours sincerely,

Xiaojing Wang

---------------

Dr. Xiaojing Wang

Professor

College of Tea Science,

Guizhou University,

Guiyang 550025, Guizhou Province, China.

Email: xjwang8@gzu.edu.cn

Round 2

Reviewer 1 Report

I did not see the authors' response to my main question, although the authors revised many.  In the abstract, "ERF might play important roles in regulating the heavy metals Cd and As stresses response by increasing the expression levels of GloS, NCED, and HIPP genes".  Is an increase in transcription level directly related to regulation?  There are many ways of regulations.  This research looked at only the level of transcripts.  At least the authors show/discuss these genes (ERF, GloS, NCED, and HIPP in terms of responding to As and Cd stress) are regulated at the transcriptional level.  Also, I was expected to see a discussion (or a model) on how ERF could play a role in As and Cd stress response in relation to GloS, NCED, and HIPP genes. 

My suggestion is the goal of this research was 'to identify candidate genes for crop development', instead.  

Author Response

Translate

Reviewer1

    I did not see the authors' response to my main question, although the authors revised many. In the abstract, "ERF might play important roles in regulating the heavy metals Cd and As stresses response by increasing the expression levels of GloS, NCED, and HIPP genes". Is an increase in transcription level directly related to regulation? There are many ways of regulations. This research looked at only the level of transcripts. At least the authors show/discuss these genes (ERF, GloS, NCED, and HIPP in terms of responding to As and Cd stress) are regulated at the transcriptional level. Also, I was expected to see a discussion (or a model) on how ERF could play a role in As and Cd stress response in relation to GloS, NCED, and HIPP genes.

    My suggestion is the goal of this research was 'to identify candidate genes for crop development', instead.

    Response1: Thank you for your comments. We are sorry for our misunderstanding of your main question. As the reviewer said, our study looked at only the level of transcripts, and we can not conclude that "ERF might play important roles in regulating the heavy metals Cd and As stresses response by increasing the expression levels of GloS, NCED, and HIPP genes". In our study, the results of WGCNA analysis revealed that the ERF transcription factor (CSS0000647) was positively correlated with five functional genes such as three GolS genes (CSS0001107, CSS0019367, and CSS0035212), one NCED (CSS0033791), and one HIPP (CSS0006162), which suggested that ERF transcription factor might also play important roles in enhancing the tolerance to Cd and As stresses. Through results of WGCNA analysis, we can not conclude that ERF could play a role in As and Cd stress response in relation to GloS, NCED, and HIPP genes. We have replaced these content with “the results of WGCNA analysis revealed that the ERF transcription factor (CSS0000647) was positively correlated with five functional genes such as three GolS genes (CSS0001107, CSS0019367, and CSS0035212), one NCED (CSS0033791), and one HIPP (CSS0006162), which suggested that ERF transcription factor might also play important roles in enhancing the tolerance to Cd and As stresses” in our manuscript.

    Response2: We also agreed with the reviewer’s opinion that the goal of our research was 'to identify candidate genes for crop development'. We have modified our manuscript according to the reviewer’s suggestion, and hope that the corrections and response answered the reviewer’s main question.

    We tried our best to improve the manuscript and made changes in our manuscript. We appreciate for the reviewers’ warm work earnestly, and hope that the corrections will meet with the approval of this journal. If you have any queries, please don’t hesitate to contact me at the address below.

Thank you and best regards.

Yours sincerely,

Xiaojing Wang

---------------

Dr. Xiaojing Wang

Professor

College of Tea Science,

Guizhou University,

Guiyang 550025, Guizhou Province, China.

Email: xjwang8@gzu.edu.cn

Reviewer 2 Report

My criticisms have been dealt with.

Author Response

Thank you very much for your valuable suggestions and comments.